# Comparison of Different Heat Treatment Processes of Selective Laser Melted 316L Steel Based on Analysis of Mechanical Properties

**DOI:** 10.3390/ma13173805

**Published:** 2020-08-28

**Authors:** Janusz Kluczyński, Lucjan Śnieżek, Krzysztof Grzelak, Artur Oziębło, Krzysztof Perkowski, Janusz Torzewski, Ireneusz Szachogłuchowicz, Krzysztof Gocman, Marcin Wachowski, Bogusz Kania

**Affiliations:** 1Institute of Robots & Machine Design, Faculty of Mechanical Engineering, Military University of Technology, 2 Gen. S. Kaliskiego St., 00-908 Warsaw 49, Poland; lucjan.sniezek@wat.edu.pl (L.Ś.); krzysztof.grzelak@wat.edu.pl (K.G.); janusz.torzewski@wat.edu.pl (J.T.); ireneusz.szachogluchowicz@wat.edu.pl (I.S.); marcin.wachowski@wat.edu.pl (M.W.); 2Department of Ceramics and Composites, Institute of Ceramics and Building Materials, 9 Postepu St., 02-676 Warsaw, Poland; artur.ozieblo@icimb.pl (A.O.); k.perkowski@icimb.pl (K.P.); 3Institute of Vehicles & Transportation, Faculty of Mechanical Engineering, Military University of Technology, 2 Gen. S. Kaliskiego St., 00-908 Warsaw 49, Poland; krzysztof.gocman@wat.edu.pl; 4Institute of Metallurgy and Materials Science, Polish Academy of Sciences, 25 Reymonta St., 30-059 Krakow, Poland; bogusz.kania@gmail.com

**Keywords:** 316L austenitic steel, selective laser melting, mechanical properties, hot isostatic pressing, precipitation heat treatment

## Abstract

In this study, we analyzed the mechanical properties of selectively laser melted (SLM) steel obtained via different modifications during and after the manufacturing process. The aim was to determine the effects of precipitation heat treatment on the mechanical properties of elements additively manufactured using three different process parameters. Some samples were additionally obtained using hot isostatic pressing (HIP), while some were treated using two different types of heat treatment and a combination of those two processes. From each manufactured sample, a part of the material was taken for structural analysis including residual stress analysis and microstructural investigations. In the second part of the research, the mechanical properties were studied to define the scleronomic hardness of the samples. Finally, tensile tests were conducted using a digital image correlation (DIC) test and fracture analysis. The treated samples were found to be significantly elongated, thus indicating the advantages of using precipitation heat treatment. Additionally, precipitation heat treatment was found to increase the porosity of samples, which was the opposite compared to HIP-treated samples.

## 1. Introduction

Additive manufacturing (AM) technology plays a significant role in various fields such as aircraft manufacturing [1], automotive manufacturing, [2] and medical applications [3,4]. The wide scope of the applications of AM technologies is of significant interest to scientists [5,6,7] because of the different material properties obtained using AM technologies in comparison to materials fabricated by conventional technologies (e.g., casting and machining). One of the most promising AM technologies is laser powder bed fusion (L-PBF), especially that dedicated to metal powder fusion. One of the most common PBF technologies is selective laser melting (SLM) which allows manufacturing parts using different types of metal powders: aluminum alloys [8,9], titanium alloys [10,11], and steel [12,13,14]. 

Significant attention has been paid on 316L steel parts processed using SLM. The application scope of this material (medicine, tube and piping technologies, food, and chemistry) is wide because of its good mechanical properties, weldability, and machinability, low cost, and very good oxidation and corrosion resistance. Because of these advantages, this material is the focus of research of many scientific facilities. One of the most important issues in additively manufactured 316L steel parts is the corrosion behavior, which is different from that of conventionally manufactured materials [15,16]. Trelewicz et al. indicated that the corrosion resistance of SLM-processed 316L steel was reduced because of the inhomogeneous solute distribution [17]. Due to the worse mechanical properties of SLM-produced elements, numerous studies are being conducted related to process parameter modification [18,19,20], in-process material annealing using laser re-melting [21,22], or post-process heat treatment [23,24]. All mentioned sources indicate the possibility of improvement of the mechanical properties. Salman et al. [25] indicated that the optimal combination of strength and ductility for the current 316L material is already reached during SLM processing and that additional heat treatments will not improve the performance of the material because the decrease of strength is not compensated by a corresponding increase of plastic deformation. 

Additionally, 316L is a base of lattice-structured elements where material behavior during different types of loading plays the most significant role. A lot of researchers have also conducted numerical analyses of lattice-structured parts [14,26,27], where it is very important to provide as many material parameters as possible to match the numerical outputs with experimental values.

Heat treatment of SLM-processed 316L steel significantly affects the mechanical properties of additive manufactured parts. Sistiaga et al. [28] determined that 316L steel after SLM processing is characterized by much higher yield and ultimate strength while keeping the high elongation close to conventionally made material. Using additional heat treatments above 950 °C resulted in no grain enlargement compared to the as-built condition. Additionally, the cellular dendritic structure was dissolved, which caused a decrease in hardness and yield strength compared to the as-built condition. Worse mechanical properties (lower hardness, ultimate tensile strength, and yield strength) after heat treatment was still higher than in conventionally made material. Molten pool boundaries between deposited layers significantly reduce material plasticity and could affect erosion growth of the material, as well as porosity [29], each molten pool is also characterized by ultrafine grain structure. Kong et al. [15] have demonstrated that it is possible to reach the same grain size for SLM processed parts after proper heat treatment in comparison with the as build SLM parts. 

Considering heat treatment, a lot of research papers are connected with hot isostatic pressing (HIP), which enables dissolving molten pool boundaries and increasing samples elongation during tensile testing. Kunz et al. [30] were able to completely dissolve a typical microstructure after SLM processing and significantly increase the plastic behavior of HIPped samples. Additionally, using HIP treatment allows keeping the preferred crystallographic orientation, which was proven by Röttger et al. [31]. In the same research paper, it was reported that HIPped parts have comparable properties with conventionally manufactured parts, which can be explained by the small grain size. 

As mentioned before, 316L has a wide scope of application, and in many cases, good tensile strength is an important requirement, while in other applications, better corrosion resistance is the most important. The main aim was of this study was to indicate how different types of process modifications, heat treatment, and its combination affect the material properties. All AM—manufactured parts were compared with conventionally made 316L steel (cold rolled sheets). To better understand how each heat treatment affects the mechanical properties of annealed samples, it is likely to use HIP treatment, solution annealing, and also a combination of those two heat treatment types, which was also suggested in [30].

## 2. Experimental

### 2.1. Material

Powder (Carpenter Additive, Widness, UK) used for the production of all samples was gas atomized steel 316L (1.4404) in an argon atmosphere. Powder particles had spherical shapes of 15–63 µm in diameter. The density of the material was 7.92 g/cm^3^, and its flowability was 14.6 s/50 g. The material chemical composition based on the quality check of the powder obtained from the supplier is shown in Table 1. 

To verify powder particles’ chemical composition additional analysis has been performed using an electron backscatter diffraction (EDS) module on a scanning electron microscope (SEM, Jeol JSM-6610, Jeol Ltd, Tokyo, Japan). Recorded data is shown in Table 2. 

The lack of other elements (Mn, P, S, and N) in the analyzed chemical composition is connected with a very low amount of the mentioned elements at the registered measurement points. 

### 2.2. Manufacturing Process 

The 3D models of testing samples were designed by SolidWorks 2019, based on the ASTM E466 96 standard, and were then used in the manufacturing process for an SLM 125HL device (SLM Solutions AG, Lubeck, Germany). All performed investigations including microstructure, residual stress, and hardness analyses were performed on the same samples to obtain reliable results. 

Our tests [32] indicated that the microhardness and porosity values changed by 10% percent after modification of selected process parameters. The mentioned parameter modifications included laser power, laser exposure velocity, and hatching distance (a gap between exposure lines). The main reason for specific parameter selection was their significant influence on energy density during the manufacturing process. In accordance with Equation (1), the energy density depends on four different parameters:(1)$$\rho_{E}=\frac{L_{P}}{e_{v}\cdot h_{d}\cdot l_{t}}$$
where *L_P_*—laser power [W], *e_v_*—exposure velocity [mm/s], *h_d_*—hatching distance [mm] and *l_t_*—layer thickness [mm].

The parameters selected for modification were affected by the device’s optical system and laser source. Layer thickness, which was not included in the research, is controlled by the worm geared mechanical system which is less precise than the galvanometric system for optic steering. The second reason for layer thickness elimination for further analysis was the lack of ability to change this parameter during a single process for each manufactured specimen. From a group of preliminarily determined parameters [32], the following three were chosen which were most significant groups (parameters are specified in Table 3):
“S_01”—the main SLM device fabrication parameters based on 316L steel.“S_17”—group of parameters which were recorded when the highest porosity was observed during the manufacturing process with the lowest energy density from all groups. In addition, the lowest microhardness was observed in samples fabricated using this group of parameters.“S_30”—group of parameters used to obtain the highest value of energy density based on a previous research [22]. Parameter selection and their descriptions were based on our own preliminary research [32,33] to clarify the interpretation of research results.


Our approach allows understanding when a particular modification (process parameters, type of heat treatment) is effective for specific applications. We can also analyze the materials’ susceptibility to different types of heat treatment. Three selected groups of parameters were used for sample production to enable structural analysis including microstructure investigation and residual stress measurement; scleronomic hardness testing and tensile tests were performed by DIC based deformation analysis. The second stage of research involved obtaining parts of samples using two different types of heat treatment–precipitation heat treatment and hot isostatic pressing on samples fabricated with two parameters groups—“S_01” and “S_17”. After heat treatment, all types of early-made tests were carried out. The last part of the research included a combination of two types of heat treatment-precipitation heat treatment after hot isostatic pressing of “S_01” and “S_17” samples. The influence of precipitation heat treatment on the samples fabricated using “S_30” parameter group was also investigated. All samples from each series were manufactured during a single process to assure the same material properties of each sample from each group. The manufacturing processes of samples was carried out in an argon atmosphere with oxygen content lower than 0.2%. All samples were oriented horizontally. The orientation assures the highest possible strength and elongation of the additively manufactured parts [34]. 

### 2.3. Research Methodology Description 

The porosity and microstructure were analyzed using a LEXT OLS 4100 confocal microscope (Olympus Corporation, Tokyo, Japan). For preparation of samples for microstructural analysis, the samples were mounted in resin, ground with abrasive papers (grade: 80, 320, 600, 1500, and 2000), and polished using a 3-μm grade diamond paste. For porosity analysis the Mountains Map software (version 6.0) was used. Acetic glycergia was used as the etchant with a composition of 6 mL HCl, 4 mL HNO_3_, 4 mL CH_3_COOH and 0.2 mL glycerol. The etching time was 40 s.

Surface residual stresses in the two main, perpendicularly oriented directions (“σ_1_, σ_2_”) based on sin^2^ψ diffractometric measurements were obtained using a D8 Discover X-ray diffractometer (Bruker Corporation, Billerica, MA, USA) with an Euler wheel and a sample positioning system along the three axes. Test samples for the research were prepared using electrical discharge machining. Radiation and beam optics were characterized by CoKα filtration. Phase analysis was performed in the CrystalImpact Match! software (version 3.0) with an ICDD PDF 4+ 2019 crystallographic database. The residual stress analysis was based on Fe 111 and Fe 311 reflections of the austenitic phases of the samples. C11 = 204.0, C12 = 133.0, and C44 = 126.0 monocrystalline elastic constants were adopted for tested 316L steel.

Sclerometric hardness measurements were conducted on CETR‘s Universal Nano and Micro Tester (CETR INC, Campbell, CA, USA), and the width of average scratches was considered. The process of material scratching involved moving the indenter. The indenter was inserted in the sample with a constant, normally oriented load and an additional constant indenter movement speed. This method of measurement allowed structural analysis of the layered material, which is a characteristic of additively manufactured materials (the perpendicular surface to the machine’s building platform). The described surface is shown in Figure 1. Sclerometric hardness was calculated using the exact differential method. For each sample, three scratches were made, and each scratch was measured three times, which is nine measurements for each sample.

Axial tensile strength tests of additively manufactured samples with structure analysis SLM of 316L steel were performed according to the ASTM E466 96 standard using a hydraulic pulsator (Instron 8802, Instron, Norwood, MA, USA). Measurements of the deformation under axial stretching were performed using an Instron 2630-112 extensometer with a measuring base of 25 mm. All samples subjected to axial tension exhibited the same geometry. Tensile tests were made accordingly to used standard in the research. Tests were run on five specimens for each configuration. For YS standard deviation was: +/− 6.1072 MPa, for UTS standard deviation was: +/− 6.1070 MPa. Sample surface deformation was analyzed during monotonic tensile tests performed using a digital image correlation (DIC) method. Deformations of the samples were observed using the Dantec Q-400 system (DANTEC DYNAMICS A/S, Skovlunde, Denmark) for three different specimen series (S_01, S_17, and S_30 before and after heat treatment) considering three characteristic parameters: yield strength, ultimate tensile strength, and breaking strength. The data received from the DIC system and tensile test machine were evaluated using ISTRA 4D software (version 4.4.1). The surface structures of the sample fractures after the tensile test were observed using a Jeol JSM-6610 SEM (JEOL Ltd, Tokyo, Japan).

Samples manufactured using the two groups of process parameters (S_01 and S_17) were obtained using hot isostatic pressing (HIP) in an argon atmosphere at 1150 °C under a pressure of 100 MPa for 3 h. The second type of heat treatment was precipitation heat treatment which was performed under an annealing temperature of 1060 °C for 6 h. To reduce the formation of high-dimensional grains, water cooling of the samples was performed directly after annealing. The second, equally important issue was to avoid the generation of the sigma phase in the material microstructure. Such precipitation is characterized by high hardness and brittleness, which negatively affect material properties.

### 2.4. Microstructural Analysis

Microscopic image observations were performed on the surface perpendicular to the machine-building platform (with visible material layers). On some sample surfaces, porosity (black, non-regular shapes) was observed. Porosity fluctuations after different types of heat treatment were observed on the microstructures and are compared in Table 4.

The most significant changes in porosity were observed after precipitation annealing of “S_17” samples, where the porosity increased from 0.879% after SLM processing to 2.41% after precipitation heat treatment. After HIP, the porosity decreased to 0.195%. After doubled heat treatment, (1st stage HIP, 2nd stage precipitation annealing) the porosity decreased to 0.01%. 

### 2.5. Residual Stress Measurements

The detailed test results of the residual stress measurements are based on 311 reflection series analysis because of the better angular position of the Fe 311 peak which is more resistant to systematic measurement errors. Residuals stress of each specimen with its orientation direction is shown in Figure 2. Stress orientation was presented according to the AM direction of samples: the vertical direction in the figure corresponds to the direction parallel to the building platform, and the horizontal direction corresponds the surface perpendicular to the building platform. All uncertainties are given with a coverage factor k = 1.

All tested samples are characterized by compressive stresses. This phenomenon if connected with research design method where it was determined the same volume of cutout in each sample to reach the most reliable results. As it is well-known, the residual stresses are strictly dependent from the part geometry. To determine how heat treatment, affect residual stresses it have to be used cutouts from the same part of each sample. 

### 2.6. Friction Force and Sclerometric Hardness Measurement

The friction force as a function of scratching length was analyzed on a surface parallel to the machine-building platform (XY plane). The main reason for data reduction was the lack of significant phenomena in the plane perpendicular to the machine’s building platform (YZ). Heat treatment completely reduced the visible influence of the layered structure in the YZ plane, which is shown in Figure 3.

Friction force during scratching registered for S_01 sample made on the sample’s layer surface (red curve in Figure 3) is characterized by a smaller friction force fluctuation than the blue one—made on this sample’s cross-section through layers surface. After HIP, no visible differences in both curves were observed. The results for the parallel plane of each sample are shown in Table 5.

Based on the recorded scratching force, the scratch dimensions, and the equation associated with the indenter type, the sclerometric hardness of the specimens can be calculated using Equation (2):(2)$$H_{Sp}=\;\frac{8\cdot F}{\pi *w^{2}}$$
where *HSp*—sclerometric hardness (Pa), *F*—normal force [N] and *w*—average scratch width [m].

### 2.7. Tensile Strength and DIC Deformation Measurements

Tensile strength results of additively manufactured specimens subjected to additional heat treatment are shown in the chart (Figure 4). The most significant growth in the tensile strength was recorded for sample S_HP17, which was subjected to two types of heat treatments. In all specimens subjected to precipitation annealing, we observed elongation and a decrease in the tensile strength. Furthermore, a conventionally manufactured sample (rolled metal sheet) was also considered (C1 in Figure 4).

Selected laser melted parts have about 40% higher YS, about 4% higher UTS, and about 30% lower elongation at failure in comparison to conventionally manufactured samples. Additional heat treatment allowed to make materials properties closer to conventionally-made. 

HIP and precipitation annealing of SLM-processed parts decrease YS and UTS strength properties with a simultaneous increase in elongation. This phenomenon is related to a significant modification of material structure, where there is a reduction of fine-grain after selective laser melting and heat treatment [15].

Strain observations using the DIC system for three different specimen series (S_01, S_17, and S_30) considering three characteristic parameters—yield strength, ultimate tensile strength, and breaking strength—were additionally compared to heat-treated equivalent parts and conventionally fabricated samples. Deformation images are shown in Figure 5. 

The results of our analysis showed that there was no evidence to demonstrate the nature of cracking during the tensile tests. The recorded SEM images, shown in Figure 6, allow us to determine the cracking type of each sample.

Arrows mark the locations of occurrence of cracks near the pores of the material. The faults visible were caused by the inter-crystalline cracking process (Figure 6).

## 3. Results and Discussion

As described in a previous study [32] and as shown in Table 3, S_17 samples showed more visible pores than other samples. The HIP process reduced the porosity of specimens manufactured using the S_17 parameter group and additional precipitation annealing after HIP caused further reductions of up to 0.01%. The same phenomenon was observed in samples manufactured using the S_01 parameters in which no significant porosity changes were observed after different types of annealing. The lack of porosity reduction after heat treatment in the S_01 samples was associated with the very low initial porosity which occurred directly after SLM processing. The same phenomenon occurred in the S_30 samples whose porosities were similar to that of S_01 samples. A significant difference in the microstructures of S_01 and S_30 samples was observed, where precipitation annealing altered the microstructure back to that of conventionally fabricated material. 

Figure 2 shows the residual stress measurement results which indicates that in all additively manufactured elements, compressive stresses were present, which could be caused by the high-temperature gradient during the manufacturing process. After HIP annealing, a significant growth (30–40%) in the residual stress was observed in the plane perpendicular to the machine’s building platform in both S_01 and S_17 samples. This could be associated with the consolidation of the material near layer borders and the additional stress generated in that area. The condition of residual stresses in samples manufactured using much higher energy densities (S_30 samples) was much lower than expected. The highest residual stress level observed was for the S_17 sample, in which the highest porosity was recorded. This level of residual stress could be associated with the porosity formation. As could be seen, precipitation heat treatment decreased the residual stresses in the S_17 samples which were characterized by higher porosity. This issue could be associated with the increase in porosity during annealing (as shown in Table 3). The opposite results were observed in the S_30 samples, where after precipitation annealing, the residual stresses increased. As could be seen in S_01 and S_17 samples that underwent HIP, additional annealing caused an increase in the residual stresses, which was caused by material shrinkage during water cooling. 

Heat treatment using precipitation annealing caused lower friction force fluctuation during surface scratching than after HIP treatment. Sclerometric hardness measurements also showed the same trend in all cases of precipitation heat treatment where this value was increased. After annealing of S_30 samples, the hardness increased. This result could be attributed to the high energy density used which reduced the grain sizes during the AM process. 

HIP treated “S_01” samples showed a total strain which was 30% higher than that of additionally manufactured specimens which were not heat treated. In addition, more significant improvement was observed in the elongation of the “S_17” samples. Elongation increased by 50%. Figure 4 shows that the trend of both curves (S_01H and S_17H) significantly approached that recorded for conventionally produced materials. However, a combination of the two heat treatment processes equalized the strength properties of the two samples (S_01 and S_17), which were characterized by extremely different properties as build parts.

The surfaces of the conventionally manufactured samples made of 316L steel (C1 in Figure 6) are characterized by plastic fracture, which is typical for such materials. For fractured surfaces of additively manufactured S_01, S_17, and S_30 samples, brittle-like cracking was observed. In specimens manufactured using the S_30 parameter group, plastic fracture with small areas of brittle-like cracking was observed. In both cases of heat treatment (HIP and precipitation), brittle plastic cracking with a local presence of inter-crystalline based setoffs was observed.

## 4. Conclusions

The data observed in this study provided the following conclusions:(1)HIP treatment significantly reduced porosity in samples manufactured using lower energy density (with high initial value of porosity in as-build samples). Using that kind of treatment in dense parts have similar effect as standard heat treatment in furnace without using additional pressure. Using HIP treatment resulted on complete removal of layered structure characterized by visible molten pools boundaries. Slow cooling affect grain size increase which was resulted on higher elongation and decreased UTS of tested samples.(2)All SLM-processed samples are characterized by compressive residual stresses, where the highest values (σ_1_ = −142 MPa and σ_2_ = −151 MPa) were registered in the “S_17” samples (manufactured using the lowest value of energy density) which was the most porous series from all tested. The lowest compressive residual stresses values (σ_1_ = −44 MPa and σ_2_ = −95 MPa) were registered in “S_30” samples (manufactured using the highest value of energy density).(3)Additional heat treatment (HIP and precipitation annealing) caused an increase in residual stress in the material. After HIP 30–40 percent increase was registered, subjecting HIPped samples additional precipitation annealing caused further residual stresses increasing also about 30–40 percent regarding state after HIP.(4)Using higher energy density cause more plastic cracking characteristic than in samples manufactured using low energy density. A similar phenomenon was observed after HIP and precipitation annealing. Using precipitation annealing only caused porosity increasing (especially in porous samples—S_17) which finally caused brittle-like cracking in that samples.

## Figures and Tables

**Figure 1 materials-13-03805-f001:**
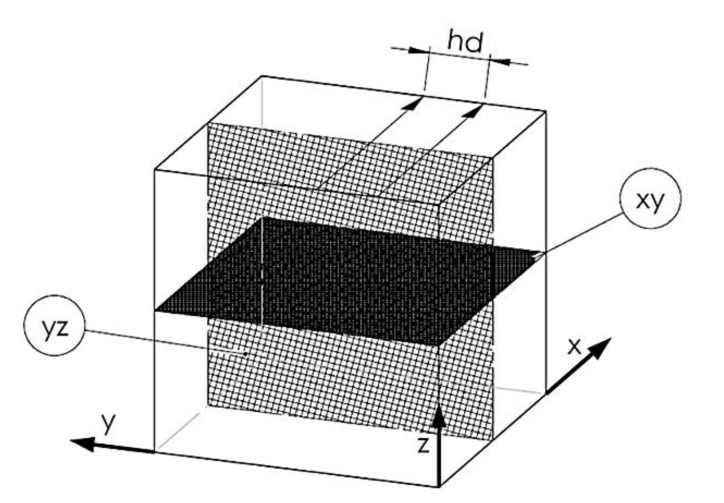
Cubic samples with the two main testing surfaces: XY: plane parallel to the building platform surface, YZ: plane perpendicular to the building platform surface, hd: hatching distance, which is the distance between the exposure lines, Z:direction of growth [32].

**Figure 2 materials-13-03805-f002:**
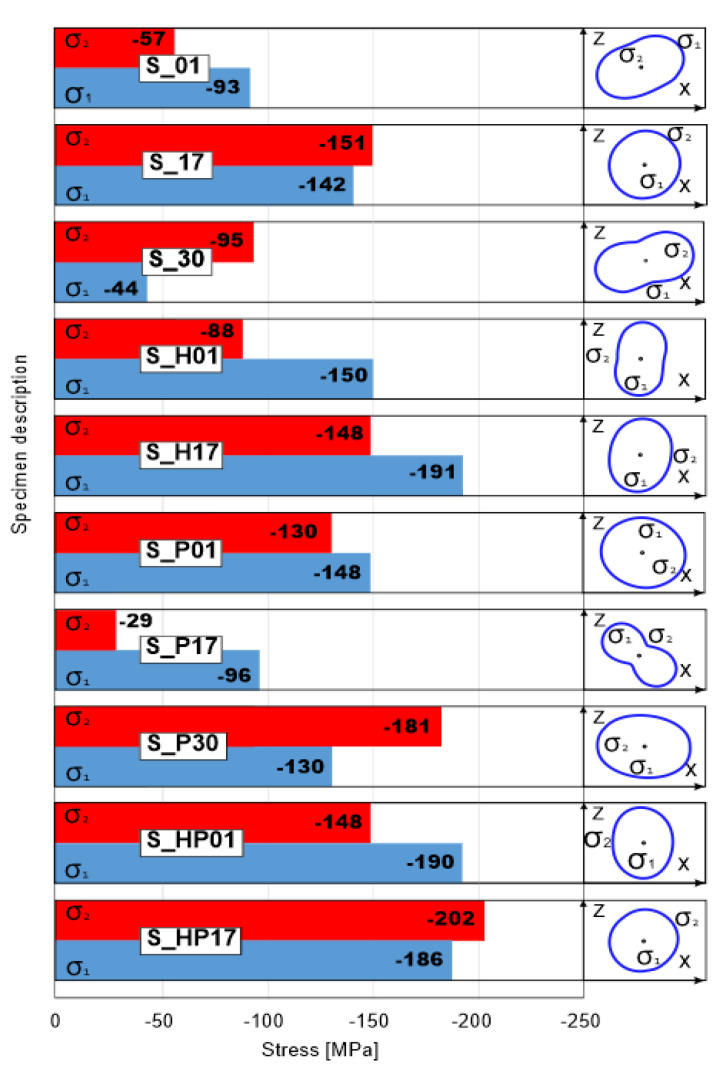
Residual stresses measured in two different directions of the samples after different types of processing (as build, underwent HIP, and precipitation annealed).

**Figure 3 materials-13-03805-f003:**
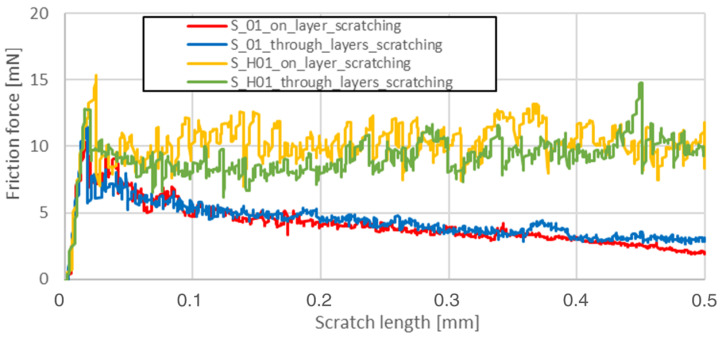
Friction forces observed while scratching the S_01 series samples before and after HIP.

**Figure 4 materials-13-03805-f004:**
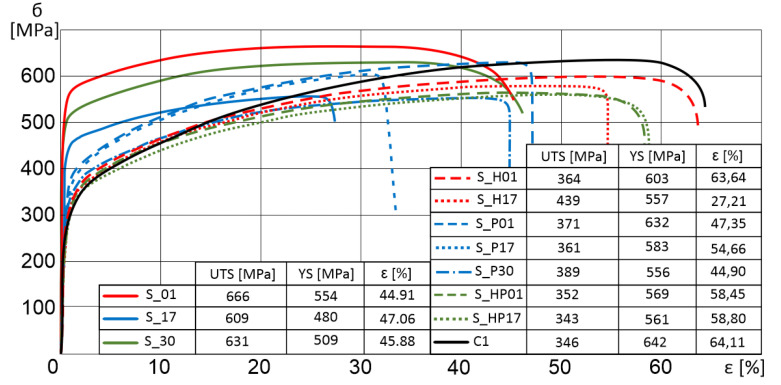
Strain-stress curves for S_01, S_17, and S_30 samples after different types of heat treatment with the additional course of conventionally made material.

**Figure 5 materials-13-03805-f005:**
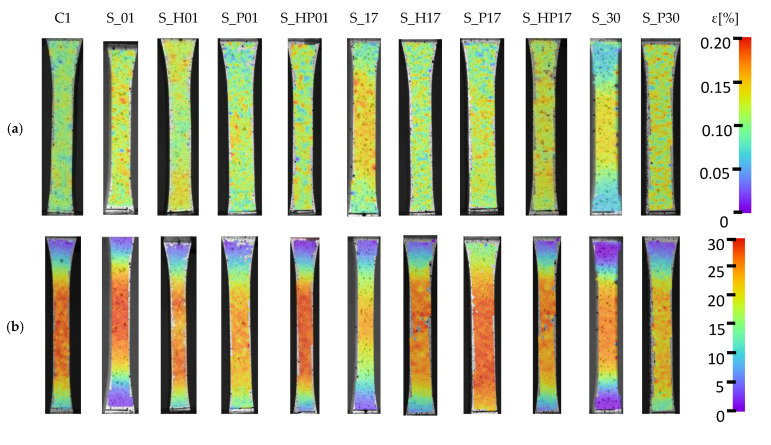
Strain distribution during monotonic tensile tests: (**a**) yield strength, (**b**) ultimate tensile strength, and (**c**) elongation at break.

**Figure 6 materials-13-03805-f006:**
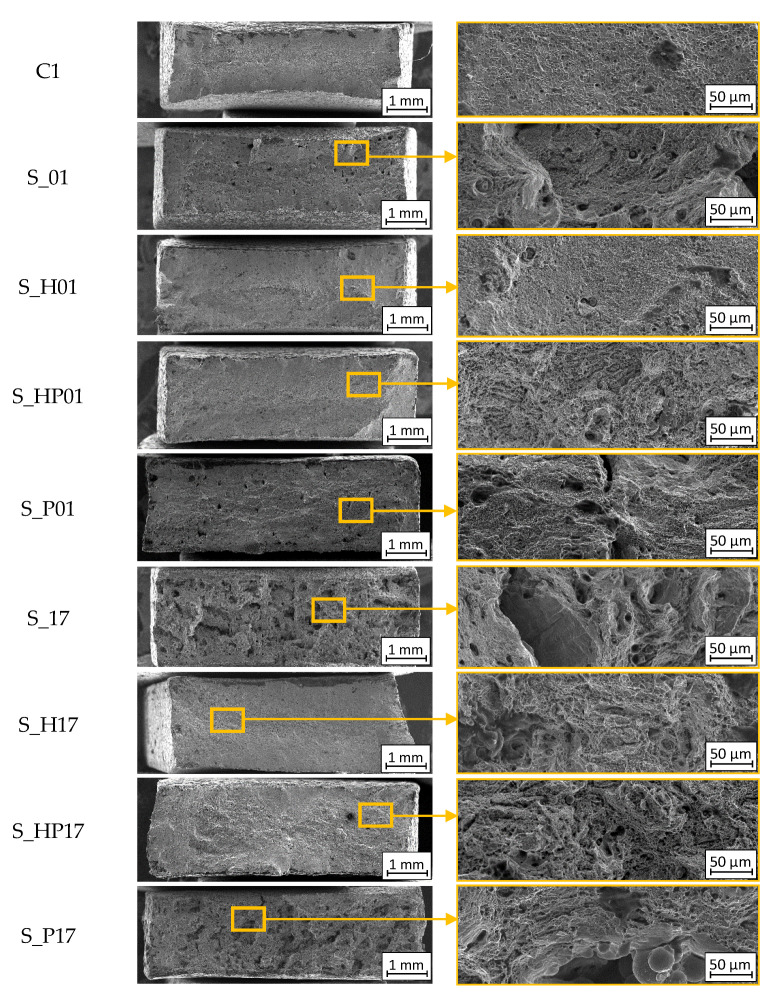
Fracture surfaces of samples after tensile tests with marked inter-crystalline cracks areas.

**Table 1 materials-13-03805-t001:** 316L steel nominal chemical composition (weight [%]).

C	Mn	Si	P	S	N	Cr	Mo	Ni
max. 0.03	max. 2.00	max. 0.75	max. 0.04	max. 0.03	max. 0.10	16.00–18.00	2.00–3.00	10.00–14.00

**Table 2 materials-13-03805-t002:** 316L steel chemical composition after SEM-EDS analysis.

Element	Apparent Concentration	Wt [%]	Wt [% Sigma]	Atomic [%]
Si	0.06	1.02	0.09	1.88
Cr	1.35	17.63	0.29	17.61
Fe	4.31	63.06	0.52	58.65
Ni	0.80	12.23	0.42	10.82
Mo	0.14	2.54	0.26	1.37

**Table 3 materials-13-03805-t003:** Parameter groups used for sample manufacturing.

Parameters Set	Laser Power L_P_ [W]	Exposure Velocity e_v_ [mm/s]	Hatching Distance h_d_ [mm]	Energy Density ρ_E_ [J/mm^3^]
S_01	190	900	0.12	58.64
S_17	180	990	0.13	46.62
S_30	120	300	0.08	166.67

**Table 4 materials-13-03805-t004:** Microstructure of S_01; S_17 and S_30 samples heat treated using different processes and their combinations.

			Parameters Set		
			S_01	S_17	S_30
a	As built	XY	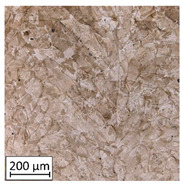	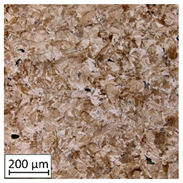	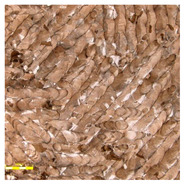
		YZ	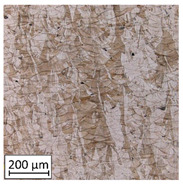	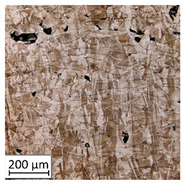	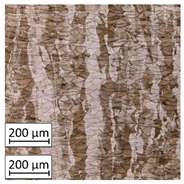
b	HIPped	XY	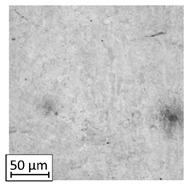	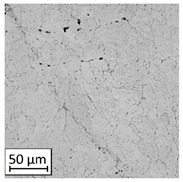	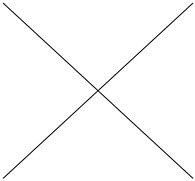
		YZ	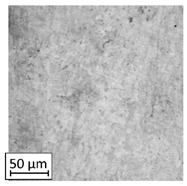	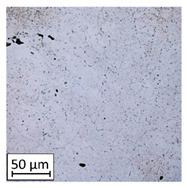	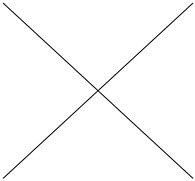
c	Precipitation annealed	XY	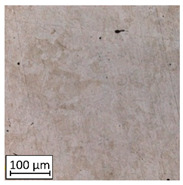	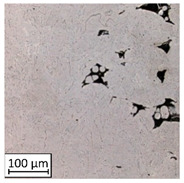	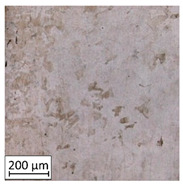
		YZ	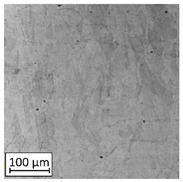	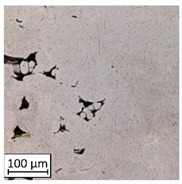	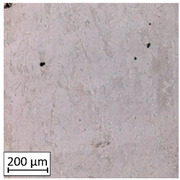
d	HIPped and precipitation annealed	XY	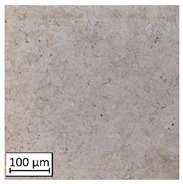	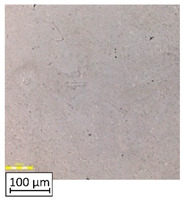	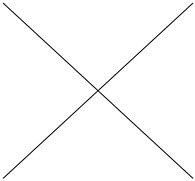
		YZ	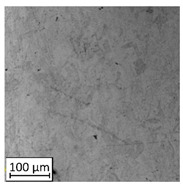	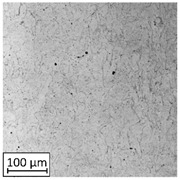	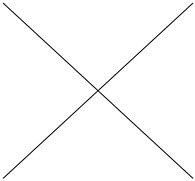

**Table 5 materials-13-03805-t005:** Friction force values and sclerometric hardness measured in XY plane of the samples after different types of processing (as build, underwent HIP, and precipitation annealed).

Measurement Type	Chart
Friction force – S_01; S_H01; S_HP01; S_P01	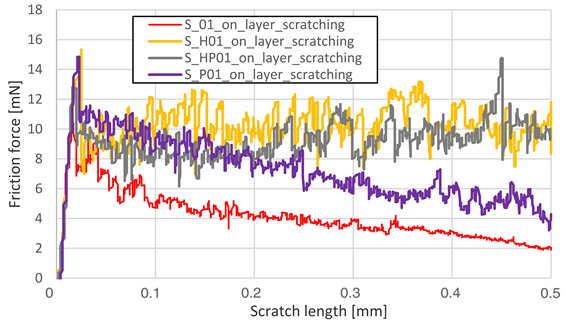
Friction force – S_17; S_H17; S_HP17; S_P17	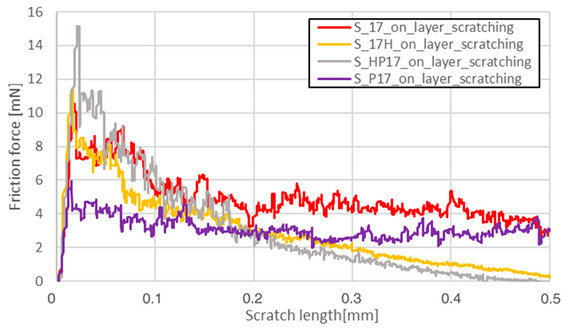
Friction force – S_30; S_P30	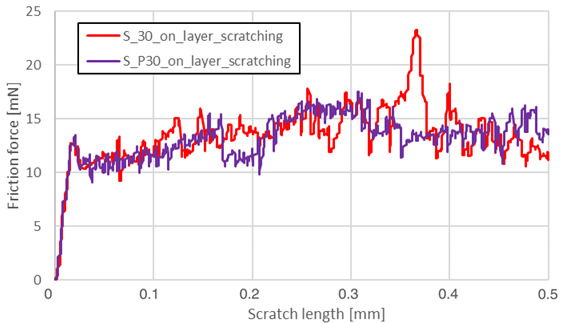
Sclerometric hardness	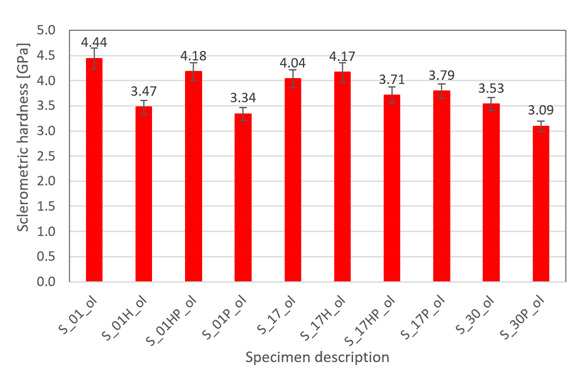

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
