# Peer review of "Comparison of Different Heat Treatment Processes of Selective Laser Melted 316L Steel Based on Analysis of Mechanical Properties"

_materials, 2020, doi:10.3390/ma13173805_

Round 1

Reviewer 1 Report

The manuscript has been well-written. The introduction need to be improved. More recent works need to be discussed and cited in the manuscript. The novelty of the manuscript is not clear and there are many similar papers are already published by other researchers in the same topic.

Regarding the experimental setup, more details need to be provided . the statistical analysis need to be explained (stats method, number of samples and ...)

Possible chemical/structural changes/variations in three samples can be discussed (phase, crstality, and possible chemical changes in their structures). XRD or Raman can be used for this purpose.

After applying my proposed comments, the manuscript can be considered for publishing.

Author Response

Dear Reviewer, 

In the beginning, we would like to thank you for reading our paper, its deep analysis, and your correction suggestions. We have attached the following corrections based on your suggestions:

  • "The introduction need to be improved. More recent works need to be discussed and cited in the manuscript. The novelty of the manuscript is not clear and there are many similar papers are already published by other researchers in the same topic."

    We improved the introduction to highlight the novelty of our paper. Please have a look at lines 74-100 (yellow-highlighted). We have also made a better literature review, and based on the following research papers:

    Sistiaga, M.L.M.; Nardone, S.; Hautfenne, C.; van Humbeeck, J. Effect of heat treatment of 316L stainless steel produced by selective laser melting (SLM). In Proceedings of the Solid Freeform Fabrication 2016: Proceedings of the 27th Annual International Solid Freeform Fabrication Symposium - An Additive Manufacturing Conference, SFF 2016; The University of Texas at Austin, 2016; pp. 558–565.

    Shifeng, W.; Shuai, L.; Qingsong, W.; Yan, C.; Sheng, Z.; Yusheng, S. Effect of molten pool boundaries on the mechanical properties of selective laser melting parts. J. Mater. Process. Technol. 2014, 214, 2660–2667.

    Kong, D.; Ni, X.; Dong, C.; Zhang, L.; Man, C.; Yao, J.; Xiao, K.; Li, X. Heat treatment effect on the microstructure and corrosion behavior of 316L stainless steel fabricated by selective laser melting for proton exchange membrane fuel cells. Electrochim. Acta 2018, 276, 293–303.

    Kunz, J.; Kaletsch, A.; Broeckmann, C. Influence of HIP post-treatment on the fatigue strength of 316l-steel produced by selective laser melting (SLM). World PM 2016 Congr. Exhib. 2016.

    Röttger, A.; Geenen, K.; Windmann, M.; Binner, F.; Theisen, W. Comparison of microstructure and mechanical properties of 316 L austenitic steel processed by selective laser melting with hot-isostatic pressed and cast material. Mater. Sci. Eng. A 2016, 678, 365–376.

  • "Regarding the experimental setup, more details need to be provided . the statistical analysis need to be explained (stats method, number of samples and ...)"

    Sclerometric hardness was calculated using the exact differential method. For each specimen, we made three scratches, and each scratch was measured three times, which is nine measurements for each specimen. We put this information also in the manuscript (lines 189-191; yellow-highlighted).

    Tensile tests were made accordingly to used standard in the research. Tests were run on five specimens for each configuration. For YS standard deviation was: +/- 6.1072 MPa, for UTS standard deviation was: +/- 6.1070 MPa. Also, this part was included in the text (lines 200-202, yellow-highlighted).

  • "Possible chemical/structural changes/variations in three samples can be discussed (phase, crstality, and possible chemical changes in their structures). XRD or Raman can be used for this purpose."

    Unfortunately, we are not able to put that type of research results because of a lack of proper equipment (XRD or Raman). We tried to put some part from the literature - basing on citations, but in our opinion using someone's job to describe our own results could be found as very unprofessional. That kind of test could significantly improve our research and we will try to find proper cooperation facilities to make it happened in our future research. 

We hope our explanation and corrections were met with your expectations. Once again thank you for your deep analysis of our manuscript.

Reviewer 2 Report

The manuscript „ Comparison of different heat treatment processes of selective laser melted 316L steel based on analysis of mechanical properties “ by Kluczyńsk et al gives a collection on material characterizations made on SLM produced austenitic steel 316L and is available as a longer version as preprint at DOI: 10.20944/preprints202004.0320.v1 or https://www.preprints.org/manuscript/202004.0320/v1.

There are several points to be clarified to allow a publication:

General:

There are several statements that are not verified by measurements (most significant microstructure/grain size) and the text/figures are not synchronized or even wrong (see below).

The conclusion should be rearranged to highlight the interesting point of the systematic correlation between energy density/HIP or annealing/residual stresses. Maybe a sorting according to the used energy density of the samples would be easier to read in the text.

Was the paper already submitted to another journal? Why was the length of manuscript shortened (length limitation at MDPI)? Seems that several figure references are not correct anymore in the text.

  1. Introduction

  1. Experimental

2.1:

- Table 1: The chemical composition is the normative reference given by the supplier (see allowed variation of the elements, e.g. Ni 10-14 weight/%). However, there is no chemical characterization of material itself.

What is the chemical composition and microstructure of the C1 sample?

Therefore, it is questionable to interpret the results, without knowledge of the real composition.

2.6:

- Sentence not understandable (p9/l238-239): “The red curve in Figure 7(?) is characterized by a smaller force fluctuation …”

Please explain what is meant by this statement. I guess that this should point to the difference of the on_layer or through_layer?

- Table 4: Correct sample name S_PH17

2.7:

- Figure 4: The figure is overloaded by information and not possible to distinguish all the lines. Further color and inset table do not correspond! Delete and set up a new one.

- Statement not correct (maybe connected to problem with figure; p11/l262):” Selective laser parts have about …” Taking the UTS for SLM and conventional there is only a 4% difference (666MPa to 642MPa).

- Statement about grain size (p12/l265): “An increased YS of SLM … attributed to the refinement in the microstructure…” This has to be verified by an analysis to allow such an assumption. There is no further assessment done in this regard. So, it is not clear what is the cause of the YS increase or if such a correlation can be drawn.

  1. Results, discussion …

As already mentioned, the conclusion should be rearranged to highlight the interesting point of the systematic correlation between energy density/HIP or annealing/residual stresses. However, wihtout further microstructur analysis the statements are not proven. The listed findings 1-4 at the end are quiet obvious and not surprising.

Figure reference does not exist (p14/l328):”.. samples made of 316L stell (P1 in Fig10) are characterized …” please align text.

Author Response

Dear Reviewer, 

In the beginning, we would like to thank you for reading our paper, its deep analysis, and your correction suggestions. We have attached the following corrections based on your suggestions:

  • "There are several statements that are not verified by measurements (most significant microstructure/grain size) and the text/figures are not synchronized or even wrong (see below)" 

    We were not able to provide some measurements connected with microstructural measurements because of a lack of proper equipment (XRD or Raman). All we were able to provide was based on microstructure observations based on literature. Regarding issues with synchronization text and figures/tables, it was checked and corrected. 

  • "The conclusion should be rearranged to highlight the interesting point of the systematic correlation between energy density/HIP or annealing/residual stresses. Maybe a sorting according to the used energy density of the samples would be easier to read in the text."

    We made the mentioned corrections and totally rebuild this part by separating conclusions as a different chapter (chapter 4). Additionally, we rephrased conclusions to make it easier to read, as you mentioned. Please have a look at lines: 351-378

  • "Was the paper already submitted to another journal? Why was the length of manuscript shortened (length limitation at MDPI)? Seems that several figure references are not correct anymore in the text."

    This manuscript has been submitted in another journal and has been rejected after peer review. We have this manuscript rebuild after the reviewer's suggestions, improve it, and prepared for publication in MDPI. 
    The length of the manuscript was shortened based on reviewers' suggestions from the mentioned journal. Indeed, several figures were removed, some were changed on tables. 

  • "Table 1: The chemical composition is the normative reference given by the supplier (see allowed variation of the elements, e.g. Ni 10-14 weight/%). However, there is no chemical characterization of material itself."

    For that kind of analysis, the most proper would be EDX analysis, but unfortunately, we were not able to make EDX analysis because of a lack of that kind of equipment. We were able to make EDS tests only, which had proven the proper share of some chemical elements. We put it below table with chemical composition and green highlighted it. 

  • "What is the chemical composition and microstructure of the C1 sample?"

    "C1" were named samples made of conventionally manufactured material used as a reference to the main research results. 

  • "Therefore, it is questionable to interpret the results, without knowledge of the real composition."

    Metallic powder obtaining by plasma atomization was made using conventionally made rods/plates, so the chemical composition in principle should be similar. Of course, it is a very good idea to make that kind of analysis but we wanted to keep the main topic of our research which is connected with additively manufactured and heat-treated parts. As we mentioned before - C1 was a reference material to extend the conclusion possibilities connected with obtained mechanical properties results. 

  • "- Sentence not understandable (p9/l238-239): “The red curve in Figure 7(?) is characterized by a smaller force fluctuation …”  Please explain what is meant by this statement. I guess that this should point to the difference of the on_layer or through_layer? "

    We rephrased the mentioned sentence: Friction force during scratching registered for S_01 sample made on the sample’s layer surface (red curve in Figure 3) is characterized by a smaller friction force fluctuation than the blue one – made on this sample’s cross-section through layers surface. - please have a look at lines 254-256.

  • "- Table 4: Correct sample name S_PH17"

    It has been corrected and green highlighted

  • "- Figure 4: The figure is overloaded by information and not possible to distinguish all the lines. Further color and inset table do not correspond! Delete and set up a new one."

    The figure has been made once again regarding your comment. 

  • "- Statement not correct (maybe connected to problem with figure; p11/l262):” Selective laser parts have about …” Taking the UTS for SLM and conventional there is only a 4% difference (666MPa to 642MPa)."

    Indeed, there was an issue with this value. We changed it to the proper value - thank you very much. 

  • "- Statement about grain size (p12/l265): “An increased YS of SLM … attributed to the refinement in the microstructure…” This has to be verified by an analysis to allow such an assumption. There is no further assessment done in this regard. So, it is not clear what is the cause of the YS increase or if such a correlation can be drawn."

    We agree with your opinion- we decided to remove this sentence completely.

  • "As already mentioned, the conclusion should be rearranged to highlight the interesting point of the systematic correlation between energy density/HIP or annealing/residual stresses. However, wihtout further microstructur analysis the statements are not proven. The listed findings 1-4 at the end are quiet obvious and not surprising.

    Figure reference does not exist (p14/l328):”.. samples made of 316L stell (P1 in Fig10) are characterized …” please align text."

    As we mentioned before, the conclusion part was completely rebuilt to highlight the most interesting parts of our manuscript. Figures numeration and their descriptions has been checked and corrected. 

    We hope our explanation and corrections were met with your expectations. Once again thank you for your deep analysis of our manuscript.

Round 2

Reviewer 2 Report

Thank you for implementing or answering to the given comments.